# Synthesis, Characterization, Cytotoxicity, Cellular Imaging, Molecular Docking, and ADMET Studies of Piperazine-Linked 1,8-Naphthalimide-Arylsulfonyl Derivatives

**DOI:** 10.3390/ijms25021069

**Published:** 2024-01-15

**Authors:** Ashanul Haque, Khalaf M. Alenezi, Ahmed Al-Otaibi, Abdulmohsen Khalaf Dhahi Alsukaibi, Ataur Rahman, Ming-Fa Hsieh, Mei-Wen Tseng, Wai-Yeung Wong

**Affiliations:** 1Department of Chemistry, College of Science, University of Ha’il, Ha’il 81451, Saudi Arabia; a.haque@uoh.edu.sa (A.H.); k.alenezi@uoh.edu.sa (K.M.A.); ahmed.alotaibi@uoh.edu.sa (A.A.-O.); a.alsukaibi@uoh.edu.sa (A.K.D.A.); 2Medical and Diagnostic Research Centre, University of Ha’il, Ha’il 55473, Saudi Arabia; 3Jamia Senior Secondary School, Jamia Millia Islamia, New Delhi 110025, India; arahman4@jmi.ac.in; 4Department of Biomedical Engineering, Chung Yuan Christian University, 200 Chung Pei Road, Chung Li District, Taoyuan City 32023, Taiwan; g11275004@cycu.edu.tw; 5Department of Applied Biology and Chemical Technology, The Hong Kong Polytechnic University, Hung Hom, Kowloon, Hong Kong, China

**Keywords:** 1,8-naphthalimide, arylsulfonyl, cellular imaging, characterization, docking, synthesis

## Abstract

To reduce the mortality and morbidity associated with cancer, new cancer theranostics are in high demand and are an emerging area of research. To achieve this goal, we report the synthesis and characterization of piperazine-linked 1,8-naphthalimide-arylsulfonyl derivatives (SA1–SA7). These compounds were synthesized in good yields following a two-step protocol and characterized using multiple analytical techniques. In vitro cytotoxicity and fluorescent cellular imaging of the compounds were assessed against non-cancerous fibroblast (3T3) and breast cancer (4T1) cell lines. Although the former study indicated the safe nature of the compounds (viability = 82–95% at 1 μg/mL), imaging studies revealed that the designed probes had good membrane permeability and could disperse in the whole cell cytoplasm. In silico studies, including molecular docking, molecular dynamics (MD) simulation, and ADME/Tox results, indicated that the compounds had the ability to target CAIX-expressing cancers. These findings suggest that piperazine-linked 1,8-naphthalimide-arylsulfonyl derivatives are potential candidates for cancer theranostics and a valuable backbone for future research.

## 1. Introduction

Accounting for nearly 8 million deaths and 14 million new cases annually, cancer remains the second leading cause of death worldwide [1]. Cancer therapy relies almost entirely on long-established technologies: surgery, radiotherapy, and chemotherapy, supplemented by more recent targeted therapies and emerging stem-cell and gene therapies [2]. Even though new potential treatments are regularly reported, the main therapeutic challenge remained largely unsolved for the past century: the complete eradication of rapidly growing malignant cells. The reasons behind our current failure to solve this challenge are manifold, including late detection, metastasis, lack of selective “magic bullet” drugs, varying responses to treatments, and the almost inevitable development of drug resistance [3]. The factor that may best improve cancer patient outcomes is earlier detection [3]. The earlier a tumor is detected and the more localized the tumor is, the better the prognosis of any therapeutic regimen [4].

1,8-Naphthalimides (I, Figure 1) are a class of π-conjugated planar molecules with wide-ranging applications [5,6]. Multiple functionalization sites, easy derivatization, and functionality-dependent chemico-biological properties are some notable features offered by this class of compounds [7]. It has been reported that 1,8-naphthalimide derivatives with two carbon-separated amines show excellent anti-tumor activity [5]. Amonafide and mitonafide (II, Figure 1) are well-known examples that impart activity via DNA intercalation and have entered clinical trials [8]. Based on this notion, various research has been carried out. For example, Kamal and coworkers [9] showed that a compound (III, Figure 1) of the type 1,8-naphthalimide-piperazine-aminobenzothiazole is a potent inhibitor of colon and lung cancers, with activity in the micromolar range. These compounds intercalate between the DNA strands and inhibit topoisomerase-II [8]. In addition, 1,8-naphthalimide derivatives have been extensively studied for the imaging and tracking of organelles such as the endoplasmic reticulum, lipid droplets, the plasma membrane, the nucleus, the Golgi apparatus, etc. [7]. Intrigued by these features and in the quest for new theranostics, we synthesized seven new 1,8-naphthalimide-arylsulfonyl derivatives (IV, Figure 1) from 1,8-naphthalic anhydride. To enhance the anticancer activity of the compounds, ethyl-separated piperazine was attached to the *N*-side of the napthalimide, followed by aryl sulfonyl group insertion. The compounds were characterized by standard analytical techniques and evaluated for in vitro toxicity against non-cancerous fibroblast (3T3) and breast cancer (4T1) cell lines. Since alkyl piperazine and aryl piperazine coumarin hybrids have been found to selectively target cancer-associated CAIX [10], which is also expressed in 4T1 cells, in silico studies (molecular docking and MD simulations) were carried out.

## 2. Results and Discussion

### 2.1. Synthesis and Characterization

In the present work, seven new piperazine-linked 1,8-naphthalimide-arylsulfonyl derivatives (SA1-SA7) were synthesized, with good yields (70–82%). All the synthesized compounds obtained were light yellow solids and stable at room temperature. All compounds were structurally characterized using ^1^H-NMR, ^13^C-NMR spectroscopy, and mass spectrometry (Appendix A), and the data are provided in the experimental section (Section 3.1). ^1^H-NMR and ^13^C-NMR showed the signals expected for aromatic, N-substituted piperazine and linker (ethyl) protons and carbons, respectively. All compounds displayed molecular ion peaks for [M + H]^+^ and/or [M + Na]^+^ peaks, supporting the structure of the final compounds.

### 2.2. Optical Studies

The ultraviolet–visible (UV-Vis) absorption and fluorescence emission spectra were measured in dilute DCM and are shown in Figure 2a, and the data are collected in Table 1. Compounds SA1–SA7 displayed high- (236–240 nm) and low-energy (~333 and 350 nm) bands in the UV region, typical for 1,8-naphthalimide derivatives [11]. The well-overlapped peaks indicate that different arylsulfonyl groups did not affect the π→π* absorption of the 1,8-napthalimide core and that there was a lack of electronic communication between two aromatic systems separated by a saturated (ethylene) linker [11,12,13]. Compared to DCM, compounds were red-shifted slightly (2–3 nm) in a more polar solvent DMSO (Appendix A). Like absorption, the emission profile of the compounds was similar (Figure 2b). When excited at 340 nm, monomeric emission based on the 1,8-napthalimide core was noted at around ~401–402 nm (Table 1), which can be ascribed to being assigned to a ^1^π–π* emitting state [11,14]. Furthermore, a low-energy band at approximately 450 nm was observed in the case of SA2 and SA5. This band could be attributed to the emission from an aggregate in the excited state that involves aromatic interactions. However, the absence of this band in the other compounds requires further investigation. The quantum yields (Φ) of the compounds were found to range between 0.030 and 0.071.

### 2.3. Biological Studies

#### 2.3.1. Cytotoxicity

For any biological application, it is crucial that molecules be non-toxic and have high cyto-compatibility. Therefore, before evaluating the cellular imaging application of SA1–SA7, the cytotoxicity was assessed against the normal fibroblast (3T3) and breast cancer (4T1) cell lines. The results of the study are provided in Figure 3 and Figure 4. When tested against normal cells, compounds SA1–SA7 showed minor toxicity (viability = 82–95% at 1 μg/mL, Figure 3). Such high cellular viability is advantageous for cell/tissue staining (depending on the specific imaging requirement) and others. Overall, the low concentration (1 μg/mL) of SA samples against non-cancerous cells confirmed that the compounds could be used for cellular imaging.

Following this, the anticancer potential of SA1–SA7 was assessed against the mouse breast cancer 4T1 cell line, which shares a close resemblance with human breast cancer in terms of growth and metastasis. The bright-field cellular images of the 4T1 cells (Figure 4) show that the cells were properly cultured to respond to three control conditions, e.g., well spread on culture well (Cont), under strong toxic stress (PC), and minor toxic stress (NC). It was noted that cells treated with the compounds led to toxic effects depending on the concentrations of the drugs (Figure 4). Even though precipitation of drugs occurs at higher concentrations (20.22 μM of SA1, Figure 4), the culture medium saturated with SA1 is believed to endow cytotoxicity. Appendix A shows the dose-viability dependence of SA1–SA7. The SA samples exerted cancer cell-killing potential differently. Among all samples, SA5 was found to be the most potent (cytotoxic) compound against 4T1 breast cancer cells (viability < 80% at 0.7 μM).

#### 2.3.2. Fluorescence Imaging

The fluorescence imaging ability of SA1–SA7 was determined on 3T3 fibroblast cells in a concentration range showing high cell viability, i.e., 1 μg/mL. As noted in the previous section, SA samples were minor/non-toxic to non-cancerous cells at the concentration of 1 μg/mL. Figure 5a–c displays the control images of both bright-field and fluorescent images. The cells co-cultured without SA samples had the typical shape of fibroblasts (Figure 5a), whereas the well-distributed cell nuclei stained with DAPI were seen yet the cellular fluorescence of the SA sample was absent (Figure 5b and Figure 5c, respectively). In contrast to control images, the cells co-cultured with SA1–SA7 were observed under a bright field (Figure 5d), via nuclei staining (Figure 5e), and with fluorescent SA4 (Figure 5f), respectively.

We noted a substantial uptake of the compounds when incubated with NIH/3T3 cells for 30 min. Compounds SA1–SA7 readily entered cells and yielded green fluorescence bioimages. At the same time, DAPI, a nuclei-staining dye, showed clear fluorescence images in the blue channel (Figure 6). An overlaid image indicated that the probes entered the cell, and fluorescence signals were localized in the perinuclear area of the cytosol, indicating that the dyes were distributed in the whole cell cytoplasm. Spindle-like cellular morphology could be seen, and varying cellular fluorescent intensity emitted was observed from different SA samples, even though the concentration of SA samples added to the cell culture medium was the same. The present study found that SA4 yielded the strongest fluorescent intensity. It is to be noted that multiple factors determine the fluorescence intensity, including the intracellular concentration of the probe, the thickness of the cultured cells, the camera constants of the optical microscope, etc. Overall, 1,8-naphthalimide-arylsulfonyl derivatives displayed both cytotoxic and fluorescent properties and hold great potential for cancer theranostics.

### 2.4. Computational Studies

#### 2.4.1. Molecular Docking

Carbonic anhydrases (CAs) have emerged as a promising druggable target among several targets [15]. CAs are ubiquitous zinc metalloenzymes present in animals, fungi, bacteria, algae, and the cytoplasm of green plants and are responsible for many biological processes, such as pH homeostasis, ion transport, respiration, etc. [16]. Several isoforms of CAs exist in humans and perform different functions [16]. Of the different types of CAs, CAIX is overexpressed in various cancerous cells (e.g., breast, colorectum, etc.). In particular, CAIX is absent in normal cells but is overexpressed in cancerous cells, including 4T1 mammary tumors [17], making it a promising target for designing and developing new agents [18]. Sulfonamides are classical inhibitors of CAs and have a high affinity towards CA enzymes, and several probes bearing sulfonamide cores are known of [19]. Moreover, several non-classical small heterocyclic inhibitors lacking sulfonamide functionality have also been developed. For example, Supuran and coworkers found an efficient inhibition of CA isoforms, especially CAIX, by rhodanine-N-carboxylate derivatives [20]. Before that study, the same group found that coumarin linked to thiazolidinone via a pyrazole linker and coumarin-linked 1,2,3-triazoles selectively target CAIX. Coumarin-alkyl piperazine and aryl piperazine hybrids for the inhibition of CAIX were also assessed [10]. Using experimental and in silico tools, Tiwari and coworkers [21] demonstrated that triazolo-pyrimidine urea derivatives exhibit excellent binding affinity towards CAIX. Therefore, we were prompted to evaluate the affinity of 1,8-naphthalimide-arylsulfonyl derivatives towards the CAIX protein.

Molecule docking of SA1-SA7 and CAIX protein (PDB: 5FL4) was carried out using the AutoDock4 tool. The docking study revealed that CAIX in complex with compound SA7 yielded the best binding affinity (−8.61 kcal/mol), followed by SA2 (−8.39 kcal/mol) > SA4 (−8.04 kcal/mol) > SA5 (−7.95 kcal/mol)~SA3 (−7.92 kcal/mol) > SA6 (−7.65 kcal/mol) > SA1 (−7.39 kcal/mol). Among other interactions, compounds SA1–SA7 formed two to five H-bonds with the receptor, involving residues Arg6, Trp9, Val130, Asn66, Arg64, His68, Gln71, Gly71, Leu91, Gln92, Ala128, Thr200, Thr201, and Pro202 (Figure 7a–d, Table 2 and Appendix A). The involvement of these residues in interaction has already been reported in both classical [22] and non-classical [21] inhibitors of CAs. To compare the results, docking was also carried out for the established CAIX inhibitor SLC-0111 (4-(3-(4-fluorophenyl)ureido)benzenesulfonamide). This compound showed a binding affinity of −8.39 kcal/mol (Appendix A).

#### 2.4.2. Molecular Dynamics (MD) Simulation

To gain further knowledge of the molecular interaction and the structural stability of the ligand–receptor pairs, MD simulations on a 100 ns scale were performed on the top four complexes containing ligands SA2, SA4, SA5, and SA7. Factors such as root mean square deviation (RMSD) and root mean square fluctuation (RMSF) were considered and compared (Table 3 and Figure 8). The RMSD value of the complexes varied in the order SA7 < SA2 < SA5 < SA4. The obtained results disclose a pattern wherein, within the initial 10 ns of the MD simulation, the ligand-bound protein complexes underwent equilibration, yielding stability over the entire 100 ns duration. Marginal instability was observed in the case of compound SA4 after 30 ns, marked by a slight deviation between 30 and 60 ns during the simulation. For compound SA7, the RMSD plot illustrated stabilization, with minor perturbations noted between 30 and 40 ns; however, the overall trajectory was consistent and stable (Figure 8a). These findings indicate that the protein–ligand complexes achieved substantial stability throughout the simulation.

The RMSD values for ligands SA2, SA4, SA5, and SA7 were calculated to be 1.82 nm, 1.40 nm, 1.66 nm, and 1.85 nm, respectively (Figure 8b). These values were obtained by tracking the conformational changes of the ligands throughout the 100 ns MD simulation. During the MD simulation, the conformational integrity of each ligand was rigorously assessed. The results unequivocally demonstrate that all ligands, namely, SA2, SA4, SA5, and SA7, maintained their structural integrity over the entire 100 ns simulation period. These results affirm the stability and robustness of the ligand structures in the complex. Negligible deviations in RMSD values further underscore the structural stability of the ligands within the protein-binding site.

The investigation of root mean square fluctuations (RMSFs) aimed to identify crucial residues involved in interactions with a ligand. The average RMSF values for compounds SA2, SA4, SA5, and SA7 were 0.81, 0.82, 0.84, and 0.81 nm, respectively. Notably, within the local domain of the CAIX protein, two substantial fluctuations were observed in the loop regions. The first notable fluctuation encompassed Asn14 and Arg19, as well as Glu79-Pro84, Glu149-Asn154, and Pro234-Leu239 residues (Figure 9). This phenomenon can be attributed to the tendency of the N and C terminals to exhibit more significant fluctuations compared to other parts of the protein. However, it is noteworthy that the compound SA7 exhibited a stable trajectory, suggesting higher protein stability. These outcomes bear significance in rational drug design and optimization, highlighting the potential of these ligands for targeted therapeutic interventions.

#### 2.4.3. Molecular Mechanics Poisson–Boltzmann Surface Area (MM/PBSA)

Molecular mechanics Poisson–Boltzmann surface area (MM/PBSA) is one of the most popular methods for predicting binding free energy (ΔG) [23]. This value offers a better understanding of the potential for complex formation than the binding affinity obtained from docking. The MM-PBSA analysis of contributing residues identifies crucial amino acids that can be instrumental in designing inhibitors. The ΔG value of SA2, SA4, SA5, and SA7 was computed for the last 20 ns (80–100 ns) of the trajectories. The g_mmpbsa tool was configured to extract snapshots from the simulated trajectories at 20 ps intervals, resulting in 1000 frames captured from each trajectory. The estimated values of the calculated ΔG were −36.78 kJ/mol, −37.93 kJ/mol, −36.61 kJ/mol, and −28.21 kJ/mol for SA2, SA4, SA5, and SA7, respectively.

#### 2.4.4. Drug Likeness and Bioavailability Studies

To underpin the drug likeness and other pharmacokinetic profiles, we estimated the absorption, distribution, metabolism, and excretion and toxicity (ADMET) of the compounds using SwissADME [24], pKCSM [25], and the Qikprop module [26,27]. The results (Table 4; Appendix A) are discussed below.

The rules of five (Ro5) is a set of rules developed by Christopher Lipinski in 1997 and provides a reasonable estimate of the drug likeness of the compounds [29]. Although various exceptions and variants to this rule exist [30,31], analysis of the features/physiochemical parameters suggested by the rules provides essential information about the molecule that can be correlated with in vitro or in vivo activities. According to the rules, a molecule with an octanol–water partition coefficient of less than 5 (log*P* ≤ 5), a molecular weight below 500 (MW ≤ 500), fewer than 10 hydrogen bond acceptors (HBA < 10), and fewer than 5 hydrogen bond donors (HBD < 5) likely act as a drug (i.e., good absorption or permeation). Later, parameters such as the topological polar surface area (TPSA ≤ 140 Ǻ^2^) and the number of rotatable bonds (RB ≤ 13 RBs) have also been added to this rule, which directly/indirectly influences drug metabolism in the human body [32,33]. A quick overview of Table 2 and Table 4 indicate that all compounds showed excellent drug-like features. Compared to the other compounds of the same series, SA2 and SA4 showed relatively lower hydrophilicity but were still within the limit (log*P* < 5). It is said that a compound that violates two or more properties might not behave like a drug [32]. However, none of the compounds displayed two or more violations. In addition, the predicted total molecular solvent accessible surface area (SASA) was found to be between 730 and 785 (Table 2) and was within the range of 300–1000. Similarly, FOSA (hydrophobic solvent accessible surface area), FISA (hydrophilic solvent accessible surface area), and PISA (carbon-pie solvent accessible surface area) were also within the defined limits (0–750, 7–330, and 0–450, respectively). On the other hand, WPSA (weakly polar solvent accessible surface area) was in the range of 0.15–0.64 for all compounds except SA4 and SA5. Despite this, it was within the allowed range (0–175). Other descriptors generated using Qikprop also supported the proposition that the compounds bear good drug-like behavior (Appendix A). For example, the octanol/water partition coefficient (QP logPo/w) for the compound ranged between 1.31 and 3.41 (recommended value: 2.0 to 6.5), whereas human serum albumin binding (QPlog Khsa) was found to be between −0.746 and 0.095 (recommended value: −1.5 to 1.5). The predicted apparent Caco-2 cell permeability was moderate to high (QP logHERG recommended value: >80% high and <25% poor). Similarly, the blood–brain partition coefficient was −0.39 to −1.72 (QP log BB, recommended value: −3.0 to 1.5), and human oral absorption was ~ 59–82% (%HOA recommended value: <25% = low, >80% high). Other descriptors were also found to be within the recommended range. It is worth noting that SA3, which showed relatively high viability against 3T3 and low toxicity against the breast cancer 4T1 cell line, showed one deviation in the RO3 during Qikprop analysis. This compound exhibited a low human oral absorption (%HOA) and a large polar surface area (PSA) compared to the other candidates. These findings align with an earlier study that suggested that descriptors such as lipophilicity, number of HBAs/HBDs, size, and rigidity are some of the crucial parameters of a heterocyclic core and determine its biological activity [34].

We also determined the absorption, distribution, metabolism, and excretion (ADME) properties of the compounds using the SwissADME [24] webtool. The results are displayed in radar (Appendix A) and Brain Or IntestinaL EstimateD permeation (BOILED-egg, Appendix A) forms. The ADME properties or bioavailability are depicted as six vertices of a pink hexagon using lipophilicity, molecular size, insaturation, fraction of sp^3^ hybridized carbons, and flexibility descriptors. Molecules falling in the radar’s pink region are considered molecules with optimum drug likeness. Except for compound SA2 (one offshoot in insatu.), there was no offshoot/deviation, and the saturation side was within the pink region. Overall, the greater the number of offshoots, the lesser the drug likeness. As is clear from the figures, among the studied physicochemical properties, there was only one insatu deviation in compound SA2, which was also evident in the Ro5 studies. In a BOILED-egg plot, a molecule falling in the white region exhibits the highest probability of being absorbed by human gastrointestinal absorption (HIA). On the other hand, those falling in the yellow region (yolk) have a high probability of permeating the brain [35]. Those falling in the grey region indicate low HIA and BB penetration. In our case, all molecules fell within the white region, and none showed BB permeation. It is also noteworthy that compound SA6 is a substrate of P-gp and effluxed by P-gp (as shown by the blue point).

#### 2.4.5. Toxicological Prediction

The toxicity profiles of the compounds were predicted using the web-based platform pkCSM, and the results are provided in Table 5 [25]. Although this was theoretically calculated and needs further experimental results to draw conclusions, some findings are noteworthy. For example, among all of the compounds, only one (SA6) showed no AMES toxicity (mutagenic) and showed good anticancer activity against 4T1 cells. Similarly, only SA2 was predicted to be the inhibitor of hERG I. Although none of the compounds possessed skin sensitization, all of the compounds were predicted to be hepatotoxic. The maximum tolerated dose for human use was highest for compound SA2. As discussed before, the compounds bore stark drug-like features and most of the descriptors were favorable (Appendix A). It was noted that the QP logHERG value of the compounds was below −5, which might be a concern. However, it should be noted that all of these values were calculated, and therefore, further in vivo studies are needed to confirm these observations.

## 3. Materials and Methods

All reagents and solvents were obtained from Sigma Aldrich (Germany) and used as received. Melting points (m.p.) were determined using an open capillary method and were uncorrected. Thin-layer chromatography was performed on silica gel-coated aluminum sheets (Merck, Germany). ^1^H- and ^13^C-NMR were collected on Bruker Spectrospin DPX 300 MHz spectrometer (Bruker Analytic GmbH, Berlin, Germany). The data were processed on MestreNova version 6.0.2–5475. The chemical shift values were recorded on the δ scale (expressed in ppm) and the coupling constants (*J*) in Hertz. Tetramethylsilane (TMS) was used as an internal standard. The following abbreviations were used for reporting spectra: s = singlet, d = doublet, dd = double doublet, m = multiplet. Mass spectra of the compounds were recorded on an Agilent 6200 series TOF/6500 series Q-TOF 10.1 (48.0) (Agilent Technologies). Absorption data were collected on a Jasco V-570 spectrophotometer (JASCO Corporation, Japan). Emission spectra were obtained on Fluoromax-4 Spectrofluorometer (HORIBA, Japan) and data were processed using FluorEssence software (V3.9). Quantum yields (at room temperature) of the compound were measured relative to a reference coumarin-153 (Φ = 0.547 in ethanol) at room temperature [36]. All QY were measured within 10% error. Φ was calculated using the following equation:$$\frac{\Phi s}{\Phi r}\;=\frac{\left(A_{r}^{\;})(\eta_{s}^{2})(I_{s}^{\;}\right)}{{(A}_{s}^{\;})(\eta_{r}^{2})(I_{r})}$$
where r and s stand for the reference and sample, respectively. *A* is the absorbance at the excitation wavelength, *η* is the index of refraction of the solvent, and *I* is the integrated luminescence intensity.

### 3.1. General Protocol for the Synthesis

The piperazine-linked 1,8-naphthalimide-arylsulfonyl derivatives (SA1–SA7) used in this study were synthesized following the protocol depicted in Figure 1. Intermediate compound 2-(2-(piperazin-1-yl)ethyl)-1H-benzo[de]isoquinoline-1,3(2*H*)-dione (3, Figure 1) was obtained by refluxing a mixture of 1,8-naphthalimide (1, 1 mmol) with 2-(piperazin-1-yl)ethan-1-amine (2, 1.1 mmol) overnight in toluene use triethylamine as a base. The final compounds (SA1–SA7) were created via a substitution reaction between substituted aryl sulfonyl chloride and (2) in DCM. The chemical composition of the compounds was determined via multiple spectroscopic techniques (vide-infra).

2-(2-(4-((2-Nitrophenyl)sulfonyl)piperazin-1-yl)ethyl)-1H-benzo[de]isoquinoline-1,3 (2H)-dione (SA1)

Yield: 0.120 g, 75%; m.p. = 170–172 °C; ^1^H-NMR (CDCl_3_) δ (ppm): δ 8.54 (dd, J = 7.3, 1.2 Hz, 2H), 8.19 (dd, J = 8.3, 1.2 Hz, 2H), 7.97–7.87 (m, 1H), 7.73 (dd, J = 8.2, 7.3 Hz, 2H), 7.69–7.64 (m, 2H), 7.59–7.55 (m, 1H), 4.28 (t, J = 6.8 Hz, 2H), 3.26 (t, J = 5.0 Hz, 4H), 2.71 (t, J = 6.8 Hz, 2H), 2.65 (t, J = 5.0 Hz, 4H). ^13^C NMR (101 MHz, CDCl_3_): δ 164.32, 148.53, 134.20, 133.78, 131.67, 131.58, 131.38, 131.08, 131.03, 128.23, 127.08, 124.14, 122.57, 55.47, 52.69, 46.10, 37.39. Calculated mass: 494.52 for C_24_H_22_N_4_O_6_S; observed mass (*m*/*z*) 495.13 [M+H]^+^.

2-(2-(4-(Naphthalen-2-ylsulfonyl)piperazin-1-yl)ethyl)-1H-benzo[de]isoquinoline-1,3 (2H)-dione (SA2)

Yield: 0.140 g, 70%; m.p. = 210–212 °C (dec.); ^1^H-NMR (CDCl_3_) δ (ppm): δ 8.46 (dd, J = 7.3, 1.2 Hz, 2H), 8.31–8.26 (m, 1H), 8.14 (dd, J = 8.3, 1.1 Hz, 2H), 7.98–7.85 (m, 3H), 7.73–7.56 (m, 5H), 4.22 (t, J = 6.8 Hz, 2H), 3.46 (s, 2H), 3.05 (s, 4H), 2.68 (d, J = 7.3 Hz, 4H). ^13^C NMR (101 MHz, CDCl_3_): δ 164.29, 134.94, 134.13, 132.63, 132.25, 131.60, 131.31, 129.36, 129.24, 129.20, 128.91, 128.15, 127.99, 127.63, 127.02, 123.22, 122.50, 55.33, 52.45, 46.26, 37.38. Calculated mass: 499.58 for C_28_H_25_N_3_O_4_S; observed mass (*m*/*z*) 500.16 [M+H]^+^.

2-(2-(4-((4-Nitrophenyl)sulfonyl)piperazin-1-yl)ethyl)-1H-benzo[de]isoquinoline-1,3 (2H)-dione (SA3)

Yield: 0.150 g, 77%; m.p. = 183–184 °C; ^1^H-NMR (CDCl_3_) δ (ppm): δ 8.50 (dd, J = 7.3, 1.2 Hz, 2H), 8.36–8.30 (m, 2H), 8.19 (dd, J = 8.3, 1.1 Hz, 2H), 7.93–7.87 (m, 2H), 7.72 (dd, J = 8.3, 7.3 Hz, 2H), 4.25 (t, J = 6.7 Hz, 2H), 3.01 (s, 4H), 2.74–2.62 (m, 6H). ^13^C NMR (101 MHz, CDCl_3_): δ 164.31, 150.18, 141.70, 134.22, 131.64, 131.32, 129.04, 128.19, 127.07, 124.37, 122.54, 55.29, 52.28, 46.21, 37.36. Calculated mass: 494.52 for C_24_H_22_N_4_O_6_S; observed mass (*m*/*z*) 495.13 [M+H]^+^.

2-(2-(4-((5-Chlorothiophen-2-yl)sulfonyl)piperazin-1-yl)ethyl)-1H-benzo[de]isoquinoline-1,3(2H)-dione (SA4)

Yield: 0.130 g, 80%; m.p. = 188–190 °C; ^1^H-NMR (CDCl_3_) δ (ppm): δ 8.54 (dd, J = 7.3, 1.1 Hz, 2H), 8.20 (dd, J = 8.3, 1.1 Hz, 2H), 7.74 (dd, J = 8.3, 7.3 Hz, 2H), 7.28–7.25 (m, 1H), 6.93 (d, J = 4.0 Hz, 1H), 4.27 (t, J = 6.8 Hz, 2H), 3.04 (t, J = 4.9 Hz, 4H), 2.75–2.66 (m, 6H). ^13^C NMR (101 MHz, CDCl_3_): δ 164.33, 137.44, 134.22, 133.85, 132.01, 131.67, 131.37, 128.22, 127.20, 127.09, 122.56, 55.31, 55.24, 46.22, 37.43. Calculated mass: 489.99 for C_22_H_20_ClN_3_O_4_S_2_; observed mass (*m*/*z*) 489.06 [M+H]+, 512.04 [M+Na]^+^.

2-(2-(4-(Thiophen-2-ylsulfonyl)piperazin-1-yl)ethyl)-1H-benzo[de]isoquinoline-1,3(2H)-dione (SA5)

Yield: 0.110 g, 73%; m.p. = 183–185 °C; ^1^H-NMR (CDCl_3_) δ (ppm): δ 8.53 (dd, J = 7.3, 1.2 Hz, 2H), 8.19 (dd, J = 8.3, 1.1 Hz, 2H), 7.72 (dd, J = 8.3, 7.3 Hz, 2H), 7.57 (dd, J = 5.1, 1.4 Hz, 1H), 7.50 (dd, J = 3.8, 1.3 Hz, 1H), 7.10 (dd, J = 5.0, 3.8 Hz, 1H), 4.27 (t, J = 6.9 Hz, 2H), 3.05 (t, J = 4.9 Hz, 4H), 2.76–2.64 (m, 6H). ^13^C NMR (101 MHz, CDCl_3_): δ 164.28, 135.92, 134.17, 132.61, 132.20, 131.67, 131.35, 128.22, 127.74, 127.06, 122.58, 77.46, 77.38, 77.14, 76.83, 55.35, 52.31, 46.22, 37.4. Calculated mass: 455.55 for C_22_H_21_ClN_3_O_4_S_2_; observed mass (*m*/*z*) 456.10 [M+H]^+^.

2-(2-(4-(Pyridin-3-ylsulfonyl)piperazin-1-yl)ethyl)-1H-benzo[de]isoquinoline-1,3(2H)-dione (SA6)

Yield: 0.135 g, 82%; m.p. = 202–204 °C; ^1^H-NMR (CDCl_3_) δ (ppm): δ 8.93 (d, J = 2.3 Hz, 1H), 8.77 (dd, J = 4.9, 1.7 Hz, 1H), 8.50 (dd, J = 7.2, 1.1 Hz, 2H), 8.17 (dd, J = 8.3, 1.1 Hz, 2H), 8.00 (dt, J = 8.0, 2.0 Hz, 1H), 7.71 (dd, J = 8.2, 7.3 Hz, 2H), 7.44 (dd, J = 8.0, 4.9 Hz, 1H), 4.24 (t, J = 6.8 Hz, 2H), 3.02 (d, J = 4.9 Hz, 4H), 2.72–2.64 (m, 6H). ^13^C NMR (101 MHz, CDCl_3_): δ 164.28, 153.43, 148.58, 135.56, 134.17, 132.58, 131.65, 131.34, 128.20, 127.05, 123.80, 122.54, 55.35, 52.31, 46.08, 37.36. Calculated mass: 450.51 for C_23_H_22_N_4_O_4_S; observed mass (*m*/*z*) 451.40 [M+H]^+^.

4-((4-(2-(1,3-Dioxo-1H-benzo[de]isoquinolin-2(3H)-yl)ethyl)piperazin-1-yl)sulfonyl)benzonitrile (SA7)

Yield: 0.140 g, 83%; m.p. = 158–160 °C; ^1^H-NMR (CDCl_3_) δ (ppm): 8.50 (dd, J = 7.3, 1.2 Hz, 2H), 8.19 (dd, J = 8.4, 1.2 Hz, 2H), 7.84–7.76 (m, 4H), 7.72 (dd, J = 8.2, 7.3 Hz, 2H), 4.24 (t, J = 6.7 Hz, 2H), 2.99 (d, J = 4.9 Hz, 4H), 2.67 (dt, J = 15.5, 5.8 Hz, 6H) ^13^C NMR (101 MHz, CDCl_3_): δ 164.28, 140.21, 134.20, 132.91, 131.65, 131.29, 128.43, 128.19, 127.07, 122.55, 117.40, 116.53, 55.30, 52.29, 46.19, 37.37. Calculated mass: 474.53 for C_25_H_22_N_4_O_4_S; observed mass (*m*/*z*) 475.25 [M+H]^+^.

### 3.2. Biological Studies

For the anticancer studies, we established a stable cell culture for mouse breast cancer cell line 4T1 (ATCC: CRL-2539) cultured in the medium composition RPMI-1640 (Gibco; Grand Island, New York, USA), 10% fetal bovine serum (Gibco, Mexico), 4.5 g/L glucose, 1.5 g/L NaHCO_3_, 1 mM sodium pyruvate (Gibco; Grand Island, New York, NY, USA), and 1% penicillin–streptomycin (10,000 U/mL). We observed typical cell morphology after the IV passage of cells unfrozen from liquid nitrogen. Then, the cells were used for the subsequent anticancer experiment. A non-cancerous fibroblast cell line (3T3/NIH, ATCC: CRL-1658) was employed for the fluorescence imaging assay. The cells were cultured in 90% Dulbecco’s modified Eagle’s medium with 4 mM L-glutamine, 2 mM sodium pyruvate, 1% non-essential amino acid, 1% penicillin/streptomycin/amphotericin, and 10% fetal bovine serum.

#### 3.2.1. Cytotoxicity Assay

A cellular viability assay was employed for the cytotoxicity of SA samples. The lower viability indicates that the samples are toxic toward 4T1 cells. A cell viability reagent (MTT, Sigma-Aldrich, St Louis, MI, USA) containing yellow tetrazolium salt (3-(4,5-dimethylthiazol-2-yl)-2,5-diphenyltetrazolium bromide) can be converted to purple formazan salt to indicate the cell viability. The 4T1 breast cancer cells were cultured at a density of 5.0 × 10^3^ cells per well in 96-well plates. The cells were treated with SA samples for 24 h. The control groups were abbreviated as Cont (100% culture medium), representing normal cell culture; PC (20% DMSO and 80% culture medium), representing toxic condition; and NC (0.1% DMSO and 99.9%), representing minimum toxic condition. MTT reagent was added to 96-well plates to measure the cytotoxic effect of the SA compounds on 4T1 cells.

#### 3.2.2. Fluorescence Imaging Assay

A total of 5.0 × 10^4^ of 3T3 cells per well in a 96-well plate were co-cultured with the 1.0 μg/mL of SA samples for 72 h. The viability of the 3T3 cells was assayed by using 10% CCK-8 reagent for 60 min to determine the toxicity of the SA samples. Afterwards, the cells were fixed with glutaraldehyde, followed by staining with 0.1 μg/mL of 4′,6-diamidino-2-phenylindole (DAPI) for the imaging of the nuclei. An inverted fluorescent optical microscope (Nikon, Eclipse Ti-S, Japan) was utilized. The fluorescent light source was Nikon Intensilight C-HGFI, and two fluorescent filter cubes (Ex = 340–380 nm, Em = 435–485 nm, corresponding to DAPI, and Ex = 465–495 nm, Em = 512–558 nm, corresponding to SA samples) were used to observe the distribution of SA samples in the cytoplasm. A scientific-grade CCD (Dhyana, model: 400 BSI, Tucsen, Fuzhou, China) was used to record the images. The magnification of the OM was 200×. The original fluorescent images were recorded in greyscale. ImageJ software (an open-source software, NIH Image, version 1.54f, Bethesda, MD, USA) was employed to merge and convert two fluorescent images into visible colors, e.g., blue for cell nuclei and green for SA1–SA7 samples.

### 3.3. Computational Details

Molecular docking was performed on a computer with a Windows 10 operating system and 32 GB RAM. MD simulations were conducted on a Linux operating system using a Dell workstation with 32 GB RAM and a ZOTAC NVIDIA RTX 3060 Twin Edge 12 GB GDDR6 graphics card.

#### 3.3.1. Docking (Ligands, Receptor, and Grid Preparation)

The chemical structure of SA1–SA7 was drawn using Marvin sketch (http://www.chemaxon.com, accessed on 29 September 2023) and energy minimization was carried out by utilizing the MM2 force field. Using the Autodock tools (ADT), Gasteiger charges were added to the ligands and saved in PDBQT format. The target protein (PDB ID: 5FL4) was downloaded from the protein databank [37]. Water molecules present in the receptor were removed, and polar hydrogen atoms were added. In the target protein, zinc (Zn^2+^) is a cofactor already present in the crystal structure of 5FL4. The partial charge was determined using AutoDock 4 before saving it in PDBQT format [38].

A grid box of size 94 × 90 × 94 Å with a 0.35 Å spacing was created to accommodate the entire protein [39]. The grid was centered at coordinates x = 6.965 Å, y = −21.786 Å, z = 57.614 Å to optimize ligand-binding orientation exploration. Default parameters for the Lamarckian genetic algorithm (LGA) were applied to generate the best molecular conformation of the molecules [40]. To assess binding poses, up to 9 conformers per ligand were considered during the docking procedure. Output files were visualized using a PyMol visualizer [41].

#### 3.3.2. Molecular Dynamics (MD) Simulation

MD simulations at 100 ns was conducted using the Desmond module (Schrödinger suite 2020) [42]. The complex, including the explicit solvent system, was investigated by utilizing the OPLS3e force field, which was integrated into the Desmond (v12.8.117 Release 2021-22) software. The molecular system underwent solvation with crystallographic water (TIP3P) molecules [43]. Periodic boundary conditions were implemented in an orthorhombic setup, extending 10 Å beyond the solute in all directions to create a buffer region. Removal of overlapping water molecules ensued, followed by neutralizing the system by adding two Na^+^ ions to achieve a concentration of 0.15 M. The Nose–Hoover thermostat and barostat ensemble (NPT) was applied to maintain the system at a constant temperature of 300 K and pressure of 1 bar, respectively. Notably, applying these techniques ensures that the simulation faithfully represents the system’s behavior in a realistic environment. A hybrid energy minimization algorithm was used to prepare the system for the MD simulations. This involved an initial stage of 1000 steps using the steepest descent algorithm and subsequent steps utilizing the conjugate gradient algorithm [44]. This approach effectively optimized the system’s energy landscape before commencing the MD simulations.

#### 3.3.3. Absorption, Distribution, Metabolism, and Excretion and Toxicity (ADMET) Studies

In silico absorption, distribution, metabolism, and excretion (ADME) and toxicity studies were carried out using the web-based online platforms SwissADME [24] and pKCSM [25]. SMILE formats (as input) of the molecules were generated using Marvin 16.11.28.0, 2016, ChemAxon (http://www.chemaxon.com, accessed on 29 September 2023).

#### 3.3.4. Determining Binding Free Energy

The molecular mechanics Poisson–Boltzmann surface area (MM/PBSA) method was employed to calculate the relative binding free energy within the protein–ligand complex [45]. MM/PBASA assesses various interaction energies, including electrostatic interactions, van der Waals interactions, polar solvation energy, and nonpolar solvation. From the final 20 ns of simulation trajectories for each protein–ligand complex, the binding free energy (ΔG) was computed.

## 4. Conclusions

In summary, we synthesized and characterized seven new piperazine-linked 1,8-naphthalimide-arylsulfonyl derivatives that are non-toxic towards normal cells but show activity against breast cancer cells. Despite the limited solubility of the compounds, they could enter the cytoplasm and impart cytotoxicity. Modeling studies predicted that compounds could act by inhibiting the CAIX enzyme, which is often expressed in different cancers. The results presented in this work support the proposition that piperazine-linked 1,8-naphthalimide-arylsulfonyl derivatives are potential candidates for cancer theranostics.

## Data Availability

Data are contained within the article.

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
