# Peer review of "Synthesis, Characterization, Cytotoxicity, Cellular Imaging, Molecular Docking, and ADMET Studies of Piperazine-Linked 1,8-Naphthalimide-Arylsulfonyl Derivatives"

_ijms, 2024, doi:10.3390/ijms25021069_

Round 1
Reviewer 1 Report
Comments and Suggestions for Authors
In this manuscript, the authors present the synthesis and characterization of seven piperazine-linked 1,8- naphthalimide-arylsulfonyl derivatives. In-vitro cytotoxicity and optical properties were included. In addition, molecular modelling was carried out to Carbonic anhydrases as a potential bioactive application, target. Unfortunately, this work in this form is not suitable for publication. Some comments below.
1. In Figure 1, for compounds SA2 and SA5, an additional peak is clearly observed at emission around 450nm (probably overlapped with a broad peak at max. 400nm.) The authors should explain, in the manuscript, what the nature of this peak is, and why it is not observed for the other compounds.
2. If the authors present the fluorescence spectra intending to characterize the optical properties of the tested compounds (as new compounds), they should also calculate, measure, the quantum yield.
3. Chapters 2.3.2 and 2.3.1 have the same title
4. The citation format is incorrect - there should be “[].” instead of “.[]”
5. The authors report the docking result as "binding energy." Unfortunately, this is not the binding energy but the binding affinity or AutoDock scoring function. And this they should use in the manuscript.
6. The authors should compare the docking results for the tested compounds with a known inhibitor, i.e., binding affinity and mode of interaction.
7. Authors should correct errors in interaction with residues, lines 166-167 and Figure 6. E.g. Gln71, Gly71, Gla71 (in the figure)????
8. The authors should describe how they validated the docking protocol.
9. The authors should describe how they optimized the ligand structure (from 2D to 3D), energy minimization, etc.
10. The authors should describe whether they have included zinc ions and how determined its partial charge
11. Authors should describe how it was calculated, prepared ligand topology, file needed for MD simulation
12. If the authors report the mean RMSD value (or any other mean value) they should also give the standard deviation.
13. In Figure 7a, in addition to the RMDS plots for the complexes, there should also be a plot of the protein without ligand.
14. The authors did not include Figure 8 in the manuscript, which they describe in Section 2.4.
15. Lines 242-250: Could the authors explain what exactly they wanted to write? I don't see much sense in these sentences. First, the authors wrote, "Root mean square fluctuations (RMSF) were investigated to identify crucial residues engaged in significant interactions with a ligand." The docking shows that ligands interact in a slightly different place than those residues given in lines 244-245.
16. The authors should also calculate free binding energy from MD simulations. These values much better characterize the possibility of complex formation than the binding affinity obtained from docking.
I hope that the authors will be able to improve their manuscript according to my comments. Therefore, I recommend major revisions. Although, I don't think it can be done in a week or 10 days. There is still a lot of work to be done to make this manuscript suitable for publication.
Reviewer 2 Report
Comments and Suggestions for Authors
Haque and colleagues prepared a manuscript on synthesis, characterization, fluorescent imaging, molecular docking and ADMET studies of piperazine linked 1,8-naphthalimide-arylsulfonyl derivatives. As stated in their abstract and introduction, their focus is synthesis, characterization and biological studies of piperazine-linked 1,8- naphthalimide-arylsulfonyl derivatives.
I recommended that this manuscript can be accepted for publication with major revision. Further efforts are required for improving the quality, grammar and coherence of the manuscript.
I have a few comments about the visual perception of this manuscript. It is not at all obvious from the entire text of the article that the results of this study can provide valuable information about the biological activity of piperazine-linked 1,8-naphthalimide-arylsulfonyl derivatives, although the authors mention this twice - in the abstract and in the introduction, but do not provide any specific information give. I would like to understand the novelty and relevance of the study, its advantages over other similar works; unfortunately, such information is not contained in the article. The conclusion also does not reveal the main result and contribution of this study.
Similar comments apply to the introduction of the manuscript, which is written very briefly and mostly contains general phrases, but does not reveal the fundamental novelty and essence of this work, why it is necessary to carry out transformations of 1,8-naphthalimide and what results of biological activity these modifications lead to. Instead, the authors provide a large number of literary references to themselves (in one sentence, a not at all obvious 6 literary references to the first author!) and casually mention the activity of compounds containing 1,8-naphthalimide
The article is also poorly structured. So, Scheme 1 and the discussion about the diagram (clause 2.2.), in my opinion, for ease of understanding, should be moved to clause 2. Results and discussion.
There are a large number of typos and grammatical and punctuation errors, e.g. Key words – 1 missing; Line 92.96 – there is no hyphen; Line 260 – extra comma; Line 262, etc. in vitro and in vivo should be written in italics; Line 355 signature to the diagram – there are no degree signs, incorrect reaction conditions (b); Acknowledgments and funding contain duplicate information, line186 via in italics; line 199 is missing a parenthesis and a space, line 350 is 3 in bold, etc.
I also have big comments about the experimental section. For compounds, only NMR spectra and mass spectra data are presented, while other characteristics (melting point, rotation angle, IR spectra) are not available. In addition, yields must be indicated not only in percentages, but also in grams; for 13C NMR spectra, it is necessary to indicate the numbering of atoms, since the number of carbons does not coincide with the gross formula, and for some reason the signals of deuterochloroform (77 ppm) are considered signals molecules, which casts doubt on whether the authors actually synthesized the compounds indicated in Scheme 1.
Round 2
Reviewer 1 Report
Comments and Suggestions for Authors
The authors addressed all my comments and revised the manuscript. I appreciate this, as it certainly took a lot of work.
I have only one comment on point 12: "standard deviation from the mean value (average))". Since the authors plotted a graph (Figure 8), I guess using Excel, that is, the RMSD values for each simulation step (time) are known. What is the problem with calculating SD? One can use Excel for this purpose.
Author Response
We would like to express our sincere gratitude to the expert reviewer for accepting our paper. We have revised the manuscript as per the reviewer's suggestion and included the SD values in Table 2.
Reviewer 2 Report
Comments and Suggestions for Authors
The authors worked and improved the manuscript. The introduction substantiates the relevance and advantage of the study in comparison with the literature, further sections and subsections are structured, appropriate conclusions are drawn, the grammar and coherence of the manuscript is improved. My recommendations for accepting this manuscript for publication
Author Response
We appreciate the reviewer's positive feedback, constructive suggestions and acceptance of our paper for publication.